# Graphene/Fe_3_O_4_ Nanocomposite as a Promising Material for Chemical Current Sources: A Theoretical Study

**DOI:** 10.3390/membranes11080642

**Published:** 2021-08-20

**Authors:** Vladislav V. Shunaev, Olga E. Glukhova

**Affiliations:** 1Department of Physics, Saratov State University, 410012 Saratov, Russia; glukhovaoe@sgu.ru; 2Institute for Bionic Technologies and Engineering, Sechenov University, 119991 Moscow, Russia

**Keywords:** graphene, iron oxide, modeling, quantum capacitance, zone structure

## Abstract

The outstanding mechanical and conductive properties of graphene and high theoretical capacity of magnetite make a composite based on these two structures a prospective material for application in flexible energy storage devices. In this study using quantum chemical methods, the influence of magnetite concentration on energetic and electronic parameters of graphene/Fe_3_O_4_ composites is estimated. It is found that the addition of magnetite to pure graphene significantly changes its zone structure and capacitive properties. By varying the concentration of Fe_3_O_4_ particles, it is possible to tune the capacity of the composite for application in hybrid and symmetric supercapacitors.

## 1. Introduction

Composites based on iron oxides and carbon nanomaterials have attracted increased attention from developers of flexible energy storage devices (lithium ion batteries and supercapacitors) [1,2,3,4]. One iron oxide, magnetite Fe_3_O_4_, is often used in materials synthesis due to its richness, ecological purity and high theoretical capacity (about 926 mA·h/g), which is almost three times higher than the capacity of graphite [5,6]. However, anodes based on this metal have low electrical conductivity and cannot maintain structural stability over a large number of charge/discharge cycles. Graphene, with its high mechanical strength and flexibility, prevents the destruction of iron oxide. In addition, the high conductivity of graphene provides this composite with high electrochemical performance.

One of the often-used G/Fe_3_O_4_ syntheses is deposition of magnetite particles into graphene oxide with subsequent removal of oxygen groups. The composite obtained in this way can be called reduced graphene oxide (rGO)/Fe_3_O_4_. Liu and Sung reported that their paper based on monodisperse Fe_3_O_4_ grown in situ on rGO sheets showed a high specific capacitance of 368 F/g at 1 A/g that remained at 245 F/g at 5 A/g after 1000 cycles, indicating its suitability as flexible anode material for supercapacitors [7]. Zhao et al. developed a novel strategy for the preparation of sandwich-structured rGO/Fe_3_O_4_ that achieved higher reversible capacity and better cycle/rate performance in comparison to bulk Fe_3_O_4_ [8]. Shi et al. showed that Fe_3_O_4_/rGO nanocomposites with mass ratio m(Fe_3_O_4_):m(rGO) = 2.8 delivered the highest specific capacitance of 480 F/g at a discharge current density of 5 A/g [9]. The assembled lithium ion capacitors based on Fe_3_O_4_, rGO and activated carbon demonstrated an outstanding energy density of 98.8 W·h/kg and power density of 3.4 kW/kg with 78.9% of capacity saved after 1000 charge/discharge cycles [10]. The rGO/Fe_3_O_4_ composites synthesized by a simple and effective low-temperature thermal annealing method exhibited an energy density of 120.0 W·h/kg, a great power density of 45.4 kW/kg (achieved at 60.5 W·h/kg) and reasonably good cycling stability, with 94.1% capacity retention after 1000 cycles and 81.4% after 10,000 cycles [11].

However, some are of the mind that decoration of graphene without oxygen-containing moieties is more attractive since oxygen functional groups increase the number of sp^2^-sp^3^ bonds that significantly reduce graphene’s conductivity [12,13]—an important parameter for chemical power sources. Taufik and R Saleh reported on a hydrothermal method of G/Fe_3_O_4_ nanocomposite synthesis [14]. Nene et al. developed a simple method to synthesize G/Fe_3_O_4_ nanocomposites in which ascorbic acid reduces Fe(acac)_3_ at a specific temperature in the presence of carboxylated graphene and ultrapure water [15]. Escusson et al. designed a negative electrode based on G/Fe_3_O_4_ obtained by ultrasonic irradiation of few-layer graphene and nanocrystalline Fe_3_O_4_ [16]. After optimization, the cell voltage equaled 1.4 V, maximal energy density 9.4 W·h/kg and maximal power density 41.1 k·W/kg. Gu and Zhu obtained G/Fe_3_O_4_ nanocomposites via a sequential freeze-drying of graphene and iron ion suspension and solvent thermal synthesis method [17]. The anode based on this nanocomposite demonstrated a high reversible capacity of ~1145 mA·h/g after 120 cycles at 100 mA/g and a remarkable rate capability of 650 at 0.5 A/g.

Attempts at G/Fe_3_O_4_ study by mathematical modeling methods should also be noted. Using the Vienna ab initio simulation package, it was shown that magnetic and electron properties in this composite are determined by different interfacial terminations between graphene and Fe_3_O_4_ [18]. DFT calculations performed using Gaussian distribution showed that the addition of magnetite to graphene increases its ionization potential, leading to the creation of new negatively charged active sites that are also ready for nucleophilic interactions [19]. Earlier authors using quantum–chemical methods showed that the growth of magnetite concentrations in the γ-Fe_2_O_3_/CNT (carbon nanotubes) film leads to increases in the amount of charge on CNT and composite quantum capacitance (QC)—one of the two total specific capacity components [20,21]. The critical review performed suggested that a similar effect could be achieved in G/Fe_3_O_4_ composites. The aim of the work is to analyze the energy and electronic characteristics of freestanding G/Fe_3_O_4_ membranes with different concentrations of magnetite that would expand our knowledge of the processes occurring in flexible energy storage with electrodes based on G/Fe_3_O_4_ composites. The objects of the study were 2D composites with mass ratios m(Fe_3_O_4_):m(G) = 1:9, 1:4, 3:7 and 1:1, since these concentrations can clearly demonstrate gradual changes in graphene capacitive properties with the addition of magnetite nanocomposites, and such composites can be synthesized in experiments [16,17,22].

## 2. Methods

The search for ground states, as well as the calculation of the studied object’s zone structure, was performed by a self-consistent-charge density-functional tight-binding method (SCC DFTB) [23]. In terms of computational speed, the SCC DFTB method is comparable to traditional semi-empirical methods but provides accuracy comparable to ab initio calculations. The method is based on the second order decomposition of total Kohn–Sham energy by charge density. The matrix elements of the undisturbed Hamiltonian $H_{\mu v}^{0}$ are represented by the minimal basis of atomic orbitals, using two particle approximation. Since the main type of interaction between graphene and magnetite particles is the van der Waals interaction, in addition to the band structure energy E_BS_, repulsive energy E_rep_ and charge fluctuation energy E_SCC_, the term E_dis_ that describes dispersion energy by Lennard-Jones potential was added:(1)$$E_{\mathrm{tot}}=E_{\mathrm{BS}}+E_{\mathrm{rep}}+E_{\mathrm{SCC}}+E_{\mathrm{dis}}.\;$$

The band energy E_BS_ is found by the formula:(2)$$E_{\mathrm{BS}}=\underset{i\mu v}{\sum }c_{\mu }^{i}c_{v}^{i}H_{\mu v}^{0},\;$$
where $c_{\mu }^{i}$ and $c_{v}^{i}$ are coefficients for the decomposition of the molecular orbital into atomic orbitals. The term E_SCC_ can be found as follows:(3)$$E_{\mathrm{SCC}}=\frac{1}{2}\underset{\alpha \beta }{\sum }\gamma_{\alpha \beta \Delta q_{\alpha }\Delta q_{\beta }}$$
where $\Delta q_{\alpha }$ and $\Delta q_{\beta }$ are fluctuations of atoms α and β, respectively; $\gamma_{\alpha \beta }$ is the function that exponentially decreases with increasing distance between the α and β atoms and directly depends on chemical hardness [24].

The basic set trans3d-0-1 was used to define the interaction between Fe, O and C atoms [25]. Optimization was performed at a temperature of 300 K with 8 × 8 × 1 Monkhorst–Pack Brillion zone sampling.

The binding energy E_b_ between graphene and Fe_3_O_4_ was found by the formula:(4)$$E_{b}=E\left(G+{\mathrm{Fe}}_{3}O_{4}\right)-E\left(G\right)-E\left({\mathrm{Fe}}_{3}O_{4}\right)$$
where $E\left(G+{\mathrm{Fe}}_{3}O_{4}\right)$ is the energy of the formed composite, and $E\left(G\right)$ and $E\left({\mathrm{Fe}}_{3}O_{4}\right)$ are the energies of isolated graphene and Fe_3_O_4_ particles, respectively.

Electron transfer between graphene and Fe_3_O_4_ nanoparticles was tracked by Mulliken population analysis [26], where the atom’s charge was calculated by the formula:(5)$$Z=Z_{A}-{\mathrm{GAP}}_{A}$$
where Z_A_ is atomic number in the periodic table and GAP_A_ is the sum of the gross orbital product over all orbitals belonging to atom A.

## 3. Results

The iron nanoparticle atomic structure had the cubic space group Fd3m as in [17]. The concentration of magnetite particles on the graphene surface was varied by the dimensions of graphene nanoparticles. Table 1 shows translation vectors L_x_ and L_y_, bonding energy E_b_, Fermi level E_f_ and relative value of transferred charge Δq(Fe_3_O_4_)/n(C) for G/Fe_3_O_4_ supercells with different mass ratios. The G/Fe_3_O_4_ atomic supercell composite with a 1:1 ratio after optimization by the SCC DFTB method is shown in Figure 1. The binding energies E_b_ of magnetite particles with graphene of different sizes calculated by Formula (1) are negative, indicating the energy benefit of the considered compounds. Binding energy is mainly contributed by changes in electronic and dispersion energies. With increases in supercell dimensions, the modulus of electronic energy rises, leading to increases in binding energy, while dispersion energy falls, which leads to decreases in binding energy. As can be seen from Table 1, the bond between the Fe_3_O_4_ particle and graphene is strongest at mass ratio 1:4 (−2.65 eV); at this ratio, the rise in electronic energy surpasses the decrease in dispersion energy. The weakest E_b_ is observed at 3:7 (−1.53 eV); at this ratio, the rise in electronic energy is lower than the decrease in dispersion energy.

At the next stage, we calculated the activation energy—this is the energy that must be expended to start rapprochement between the components. For this purpose, we have constructed a graph of the dependence of dispersion energy $E_{\mathrm{dis}}^{’}=E_{\mathrm{dis}}-E_{\mathrm{dis}}^{\min}$ on the distance between graphene and iron particles, where $E_{\mathrm{dis}}$ is the value of dispersion energy at the current mutual arrangement of objects, and $E_{\mathrm{dis}}^{\min}$ is the minimum value of dispersion energy in the considered interval from 0 to 2 Å (Figure 2). Herewith, the value 0 Å corresponds to the equilibrium distance between graphene and the Fe_3_O_4_ particle after optimization by the SCC DFTB method. The activation energy was assumed as the height of the resulting potential barrier, which must be overcome by a magnetite particle to attach to the graphene surface. As can be seen from Figure 2, the highest activation energy must be spent in the case of the minimum concentration of iron particles (1:9 (0.21 eV)), and the lowest in the case of 3:7 (0.07 eV). Note that the main part of the activation energy is spent on the curvature of the graphene sheet in the area of contact with the magnetite particle (Figure 1b). It is this mutual arrangement that provides the greatest binding energy.

DOS curves for the considered structures are shown in Figure 3. It is notable that the doping of graphene with magnetite particles shifts the Fermi level to zero due to the presence of graphene oxide, as in the case of CNT doping with magnetite particles [21]. The value of the Fermi level for pure graphene is −4.68 eV; the values for G/Fe_3_O_4_ nanocomposites are shown in Table 1. It is seen that the DOS curve for pure graphene is close to zero at the Fermi level. The value of DOS rises with the growth of magnetite concentration and reaches maximum at the 1:1 ratio. In addition, it can be seen from the presented graph that at the mass ratio 1:9, the local minimum at the Fermi level as well as the symmetry-of-curve relative to the Fermi level line is observed, but that with a further increase in the concentration of magnetite, the local minimum and symmetry disappear; this indicates significant changes in the electronic properties of the material. The local maxima of DOS curves in the region of −6.8 to 2.0 eV are observed for all the considered supercells. In the region near −7 eV, the DOS values are located in the range from 0.46 eV^−1^ for 1:1 to 0.60 eV^−1^ for pure graphene; in the region near 2 eV, the DOS values are located in the range from 0.41 eV^−1^ for 1:4 to 1:9 to 0.42 eV^−1^ for 1:1. Note that after joining the graphene surface, the magnetite particles act as donors and transfer part of their charge to graphene, which becomes electronegative. The value of the total transferred charge varies between 1.41 and 1.74 e. However, the amount of charge per number of carbon atoms in graphene increases markedly with increases in the mass fraction of magnetite (Table 1).

Figure 4 shows the dependence of the QC on the voltages of pure graphene and graphene with different concentrations of Fe_3_O_4_ iron particles on its surface. As we can see, when the mass ratio of iron oxide to graphene is 1:9, the minimum QC shifts from 0 to −0.2 V, which is typical for graphene in the presence of various impurity defects [27,28,29]. A further increase in the concentration of iron oxide leads to a significant violation of the symmetry of the QC curve relative to the local minimum. An increase in the mass fraction of magnetite leads to a noticeable increase in the value of the QC at 0 V from 36.46 F/g for pure graphene to 583.52 F/g in the case of m(Fe_3_O_4_):m(G) = 1:1. This can be explained by significant increases in the relative charge q being transferred from magnetite to graphene.

The dotted lines in Figure 4 indicate the voltages corresponding to the stability limits of electrolyte systems with water (H_2_O) and propylene carbonate (PC), which are often used in such systems as a solvent. The values of the QC at these boundaries relative to the values at 0 V C_Q_-C_Q_ (0) are presented in Table 2. This table allows us to estimate the contribution of the Faraday and non-Faraday components in the accumulated or given charge, depending on the applied voltage. For pure graphene, the Faraday component of the capacity, determined by QC, increases markedly when the voltage deviates from 0 V. For G/Fe_3_O_4_ nanocomposites, the Faraday component decreases when the reduction potential of H_2_O is reached. Beneath the PC oxidation potential value, the QC changes take almost the same values for all the considered objects.

## 4. Conclusions

Using the SCC DFTB method, G/Fe_3_O_4_ atomic supercell composites were obtained with mass ratios m(Fe_3_O_4_):m(G) = 1:9, 1:4, 3:7 and 1:1, observed during experimental synthesis. The energy characteristics of the studied objects indicate that the formation of 3:7 composites is the least energy-consuming, while the 1:4 compound is the most stable. These results can be used in experiments that receive G/Fe_3_O_4_ composites without preliminary treatment of GO—for example, by ultrasonic irradiation [16] or hydrothermal method [14]. Increases in the concentration of magnetite particles lead to significant changes in the zone structures of composites, in particular, the shift of the Fermi level to the right. The growth of the mass fraction of magnetite leads to a noticeable increase in the value of the quantum capacitance at 0 V, from 36.46 F/g for pure graphene to 583.52 F/g in the case of 1:1. Thus, the improvement in the capacity properties of G/Fe_3_O_4_ composites with increases in the proportion of Fe_3_O_4_, is mainly caused by significant increases in the Faraday component, due to the participation of Fe_3_O_4_ in the electrochemical process. These observations may be useful during the design of electrodes for flexible storage devices based on G/Fe_3_O_4_ composites [7,8,9,10,11,16,17]. The addition of magnetite particles to graphene leads to the appearance of asymmetry in the quantum capacitance of composites. Thus, by varying the concentration of magnetite, these electrode materials can be used in both hybrid and symmetric supercapacitors [29].

## Figures and Tables

**Figure 1 membranes-11-00642-f001:**
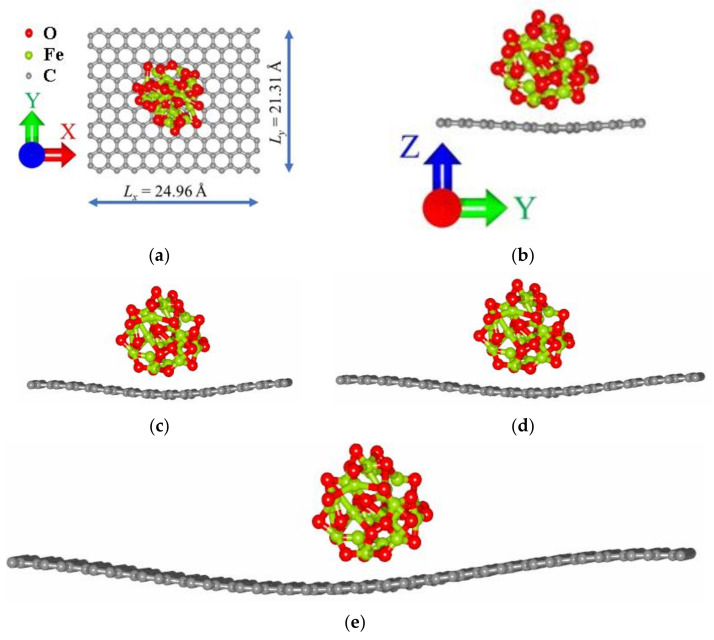
The atomic supercell of the membranes G/Fe_3_O_4_ after optimization by the SCC DFTB method, with mass ratios: (**a**) 1:1 top view; (**b**) 1:1 side view with translation vectors L_x_ and L_y_; (**c**) 3:7 side view; (**d**) 1:4 side view; (**e**) 1:9 side view.

**Figure 2 membranes-11-00642-f002:**
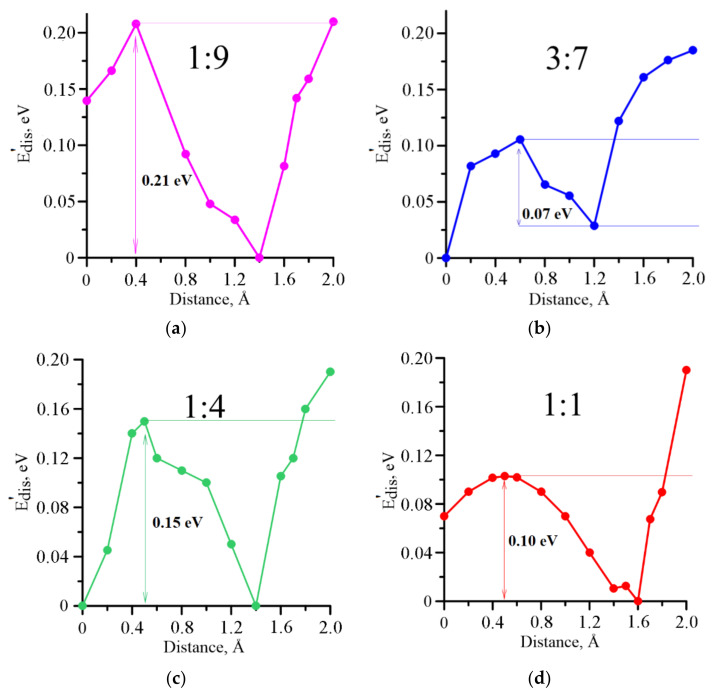
The dependence of dispersion energy on the distance between graphene and the Fe_3_O_4_ particle for different mass ratios of m(Fe_3_O_4_):m(G): (**a**) 1:9; (**b**) 3:7; (**c**) 1:4; (**d**) 1:1.

**Figure 3 membranes-11-00642-f003:**
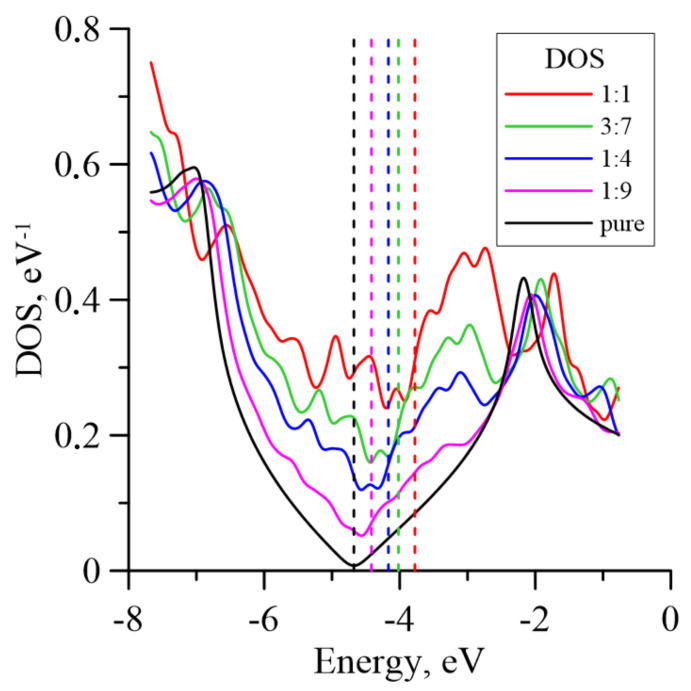
DOS curves for pure graphene and G/Fe_3_O_4_ membrane nanofilms with different concentrations of magnetite particles. The dotted lines correspond to Fermi levels.

**Figure 4 membranes-11-00642-f004:**
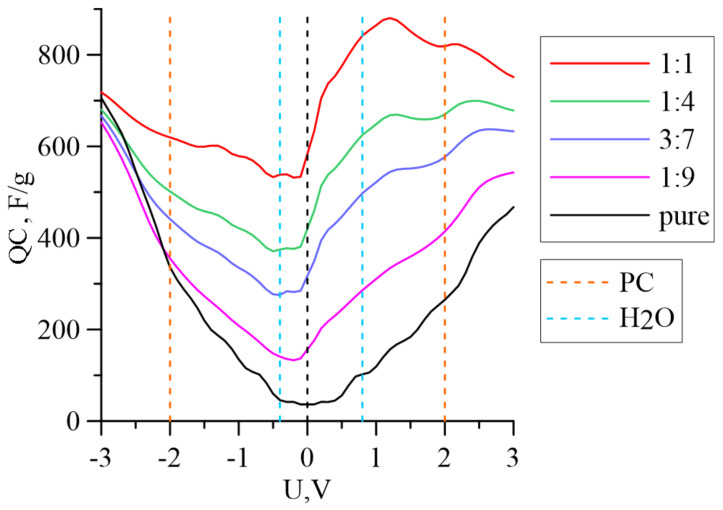
Dependence of QC for pure graphene and G/Fe_3_O_4_ membranes with different concentrations of magnetite particles on applied voltage. The black dotted line corresponds to 0 V.

**Table 1 membranes-11-00642-t001:** Translation vectors L_x_ and L_y_, bonding energy E_b_, Fermi level E_f_ and relative value of transferred charge Δq(Fe_3_O_4_)/n(C) for G/Fe_3_O_4_ supercells with different mass ratios.

m(Fe_3_O_4_):m(G)	L_x_, Å	L_y_, Å	E_b_, eV	E_f_, eV	Δq(Fe_3_O_4_)/n(C), me
1:9	62.41	59.63	−2.36	−4.42	1.15
1:4	37.44	38.34	−2.65	−4.27	3.23
3:7	32.45	29.80	−1.53	−4.02	4.29
1:1	24.96	21.31	−1.99	−3.78	7.10

**Table 2 membranes-11-00642-t002:** Values of QC for pure graphene and G/Fe_3_O_4_ membranes with different concentrations of magnetite particles at the stability limits of electrolyte systems with water and propylene carbonate relative to values at 0 V C_Q_-C_Q_ (0).

m(Fe_3_O_4_):m(G)	−0.4 V (red. H_2_O)	0.8 V (ox. H_2_O)	−2.0 V (red. PC)	2.0 V (ox. PC)
Pure graphene	45.8	102.2	335.8	265.0
1:9	−15.8	129.2	198.6	257.3
1:4	−41.5	180.7	123.9	259.3
3:7	−45.6	204.6	81.1	249.4
1:1	−36.4	267.2	44.8	243.8

## Data Availability

Not applicable.

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
