# Peer review of "Graphene/Fe3O4 Nanocomposite as a Promising Material for Chemical Current Sources: A Theoretical Study"

_membranes, 2021, doi:10.3390/membranes11080642_

Round 1

Reviewer 1 Report

In this manuscript, the authors use the self-consistent charge density functional tight-binding method (SCC DFTB) to search the ground states and calculate the studied objects zone structure. The result shows that the addition of magnetite to pure graphene significantly changes its zone structure and capacitive properties. By varying the concentration of Fe3O4 particles it’s possible to tune capacity of the composite for application in hybrid and symmetric supercapacitors. Overall, it is helpful for researchers who are interested in this field. However, it is potentially publishable once the authors have addressed the following questions:

  1. The author should mark the different atoms represented by the different colored spheres in Figure 1.
  2. In the last sentence of page 2, the author mentioned that bonding energy Eb is the strongest at the mass ratio 1:4 (-2.65 eV), the least strong – at 3:7 (-1.53 eV). Here, the author should explain why the maximum and minimum values appear in these ratios.
  3. The author should put the four curves with different mass ratios in Figure 2 into one graph to make it easier to compare the differences between the curves.
  4. The author mentioned on page 3 that the main part of the activation energy is spent on the curvature of the graphene sheet in the area of contact with the magnetite particle, so the atomic supercell of the nanofilm G/Fe3O4 with four mass ratios should be placed in Figure 1.
  5. Graphene plasmon may also be contributed to the properties in the manuscript, see Reviews in Physics, 2021, 6, 100054.
  6. The three curves with mass ratios of 1:1, 3:7, and 1:4 in Figure 3 need more explanation.

Author Response

We thank Referee for his time that have been devoted to our paper. Here are answers on his notices.

1.The author should mark the different atoms represented by the different colored spheres in Figure 1.

The different atoms were marked in Fig. 1.

2. In the last sentence of page 2, the author mentioned that bonding energy Eb is the strongest at the mass ratio 1:4 (-2.65 eV), the least strong – at 3:7 (-1.53 eV). Here, the author should explain why the maximum and minimum values appear in these ratios.

The detailed attempt of explanation for these results was added:

«The binding energy is mainly contributed by changes in electronic and dispersion energies. With increase of the supercell dimensions the modulus of electronic energy rises that leads to increase of binding energy while the dispersion energy falls that leads to decrease of binding energy. As can be seen from Table 1, the bond between the Fe3O4 particle and graphene is the strongest at the mass ratio 1:4 (-2.65 eV): at this ratio the rise of electronic energy surpasses the decrease of dispersion energy. The least strong Eb is observed at 3:7 (-1.53 eV): at this ratio the rise of electronic energy is lower the decrease of dispersion energy.»

3 .The author should put the four curves with different mass ratios in Figure 2 into one graph to make it easier to compare the differences between the curves.

 Thanks to Reviewer for the advice for improving the article. We’ve tried to place all curve at one figure (the result is in attached file). To our mind, this figure demonstrates the nature of processes not clear as it was before. But we can put this figure except previous Fig. 2 if Reviewer end Editor desire.

4. The author mentioned on page 3 that the main part of the activation energy is spent on the curvature of the graphene sheet in the area of contact with the magnetite particle, so the atomic supercell of the nanofilm G/Fe3O4 with four mass ratios should be placed in Figure 1.

The atomic supercell of the nanofilm G/Fe3O4 with four mass ratios were placed in Figure 1

5. Graphene plasmon may also be contributed to the properties in the manuscript, see Reviews in Physics, 2021, 6, 100054.

Thanks for very interesting paper. We don’t know how to include this paper to the References since influence of graphene plasmons is not consider here.

6. The three curves with mass ratios of 1:1, 3:7, and 1:4 in Figure 3 need more explanation.

The explanation of curves in Fig.3 was expanded.

«It’s seen that DOS curve for pure graphene is close to zero at the Fermi level. The value of DOS rises with the growth of magnetite concentration and reaches maximum at the 1:1 ratio. Also, it can be seen from the presented graph that at the mass ratio 1:9 the local minimum at the Fermi level as well as the symmetry of curve relative to Fermi level line is observed, but with a further increase in the concentration of magnetite the local minimum and symmetry disappear that indicates significant changes in the electronic properties of the material. The local maxima of DOS curves in the region of -7 and 2 eV are observed for all the considered supercells. In the region near -7 eV the DOS values are located in the range from 0.46 eV-1 for 1:1 to 0.60 eV-1 for pure graphene; in the region near 2 eV the DOS values are located in the range from 0.41 eV-1 for 1:4 and 1:9 to 0.42 eV-1 for 1:1.»

Reviewer 2 Report

The work entitled Graphene/Fe3O4 nanocomposite as a promising material for chemical current sources: a theoretical study Shunaev and Glukhova addresses theoretical calculations of graphene/Fe3O4 nanocomposites. I believe that the general interest in this topic is in line with submission to Membranes journal and potentially of broad interest.

 However, the manuscript in the present form suffers from the required novelty and potential impact to be published. Consequently, I recommend to publish this work after some major changes.

  1. The title of the paper is misleading since the promising nanocomposite for supercapacitors and batteries published in the literature is graphene oxide/Fe3O4 nanocomposite instead of graphene/Fe3O4.
  2. The authors perform calculations of 2D-composites with mass ratios m(Fe3O4):m(G) = 1:9, 1:4, 3:7 и 1:1, since such nanocomposites were synthesized in experiment of reference [4]. The striking omission of this work is not consider graphene oxide for the calculations. In many of the references the authors used in the introduction, those works refer to graphene oxide or reduced graphene oxide as material able to avoid clustering of Fe3O4 nanoparticles. The authors should consider how realistic are the systems they simulated with the experimental systems published, or alternatively, how interesting is the graphene/Fe3O4 system for supercapacitors and batteries.

Minor comments:

  • The following sentence should be revised: “In addition, the superconductivity of graphene provides this composite with high electrochemical performance”. The authors should consider to change superconductivity by high conductivity or similar.

In conclusion, considering all the comments above, I am inclined to reject the present work for publication in Membranes journal.

Author Response

The work entitled Graphene/Fe3O4 nanocomposite as a promising material for chemical current sources: a theoretical study Shunaev and Glukhova addresses theoretical calculations of graphene/Fe3O4 nanocomposites. I believe that the general interest in this topic is in line with submission to Membranes journal and potentially of broad interest.

  1. The title of the paper is misleading since the promising nanocomposite for supercapacitors and batteries published in the literature is graphene oxide/Fe3O4 nanocomposite instead of graphene/Fe3O4.
  2. The authors perform calculations of 2D-composites with mass ratios m(Fe3O4):m(G) = 1:9, 1:4, 3:7 и 1:1, since such nanocomposites were synthesized in experiment of reference [4]. The striking omission of this work is not consider graphene oxide for the calculations. In many of the references the authors used in the introduction, those works refer to graphene oxide or reduced graphene oxide as material able to avoid clustering of Fe3O4 nanoparticles. The authors should consider how realistic are the systems they simulated with the experimental systems published, or alternatively, how interesting is the graphene/Fe3O4 system for supercapacitors and batteries.

Thanks for attentive reading of our manuscript!

We’re sorry for the entanglement. Really, in initial version of manuscript there were several papers addressed to the composite graphene oxide/Fe3O4. In the new version these links were removed from references. The introduction was carefully revised. There is a block that is devoted to rGO/Fe3O4 composites and the block devoted to G/Fe3O4 that was obtained without preliminary GO synthesis.

However, there is a mind that decorating of graphene without oxygen containing moieties is more attractive since oxygen functional group increase the number of sp2-sp3 bonds that significantly reduces graphene’s conductivity [12, 13] – important parameter for chemical power sources. Taufik and R Saleh reported about hydrothermal method of the G/Fe3O4 nanocomposite synthesis [14]. Nene et al. developed the simple to synthesize G/Fe3O4 nanocomposite in which ascorbic acid reduces Fe(acac)3 at a specific temperature in presence of carboxylated graphene and ultrapure water [15]. Escusson et al. designed negative electrode on the base of G/Fe3O4 obtained by ultrasonic irradiation of the few-layer graphene and nanocrystalline Fe3O4 [16] After optimization this cell voltage equaled to 1.4 V, maximal energy density – to 9.4 W·h/kg and the maximal power density – to 41.1 k·W/kg. Gu and Zhu obtained G/Fe3O4 nanocomposite via a freeze-drying of graphene and iron ion suspension and by a solvent thermal synthesis method sequentially [17]. The anode on the base of this nanocomposite demonstrated a high reversible capacity of ~1145 mA·h/g after 120 cycles at 100 mA/g and a remarkable rate capability of 650 at 0.5 A/g.

The papers 12, 13, 15, 16 and 17 were added to the References. So the systems on the base of graphene and Fe3O4 are realistic and already used as the materials for chemical power sources.

The following sentence should be revised: “In addition, the superconductivity of graphene provides this composite with high electrochemical performance”. The authors should consider to change superconductivity by high conductivity or similar.

 That was fixed.

Reviewer 3 Report

In this manuscript, the authors performed a theoretical investigation through DFT calculations for the Graphene/Fe3O4 nanocomposites with different mass ratios. It may be important for researchers in the field of energy storage devices. On the whole, the manuscript is well-prepared and the results are clearly described. However, there are still some problems needing to be solved. The detailed comments are as follows:

1) In the Introduction, the authors enumerate many application examples of the G/ Fe3O4, which seems tedious. It may be better to delete some of them, and underline the characteristics and highlights of each of the other examples.

2) Some application examples of Graphene/Fe3O4 nanocomposite in batteries should also be included in the Introduction.

3) More method description is needed.

4) When discussing the results, some relating experimental application examples, if found from other literature, are suggested to be added, so that this paper looks more like an academic article than a simple data report. 

Author Response

In this manuscript, the authors performed a theoretical investigation through DFT calculations for the Graphene/Fe3O4 nanocomposites with different mass ratios. It may be important for researchers in the field of energy storage devices.

Thank you for this opinion!

On the whole, the manuscript is well-prepared and the results are clearly described. However, there are still some problems needing to be solved. The detailed comments are as follows:

  • In the Introduction, the authors enumerate many application examples of the G/ Fe3O4, which seems tedious. It may be better to delete some of them, and underline the characteristics and highlights of each of the other examples.

We partially agree with Reviewer but there is the problem that two other Referees asked to expand Introduction and even to add new modern papers. Also the Introduction was revised; we hope that new version will be not so «tedious».

  • Some application examples of Graphene/Fe3O4nanocomposite in batteries should also be included in the Introduction.

The links 2 and 17 in new version correspond to application of G/Fe3O4 in batteries.

  • More method description is needed.

The Chapter 2 Methods was expanded.

  • When discussing the results, some relating experimental application examples, if found from other literature, are suggested to be added, so that this paper looks more like an academic article than a simple data report. 

The following sentence was added to Conclusion.

These result can be used in experiments that receive composites G/Fe3O4 without preliminary treatment of GO – for example, by ultrasonic irradiation [16] or hydrothermal method [14].

These observarions may be useful during design the electrodes for flexible storage devices on the base of G/Fe3O4 nanofilms [7-11, 16, 17].

Reviewer 4 Report

The authors report very interesting theoretical study about the effect of formation of iron oxide and graphene hybrids on the mechanical and conductive properties of produced composite and its theoretical capacity for energy storage devices. They reached the conclusion through studying various mass ratio of iron and graphene. They found the addition of magnetite to pure graphene significantly changes its zone structure and capacitive properties. I am sure that this study will help in understanding and developing new composites for energy storage in the future. The manuscript can be considered for publications in Membranes after tackling out these two comments:

  1. The authors should convey more recent literatures on Iron/Graphene and/or carbon materials in the introduction for superconductors and lithium-ion batteries. For example: https://www.mdpi.com/2079-4991/9/5/776
  1. I wonder if the authors can study more samples with different Fe3O4/G mass ratio, e.g. 3:19 and 4:1.

Author Response

The authors report very interesting theoretical study about the effect of formation of iron oxide and graphene hybrids on the mechanical and conductive properties of produced composite and its theoretical capacity for energy storage devices. They reached the conclusion through studying various mass ratio of iron and graphene. They found the addition of magnetite to pure graphene significantly changes its zone structure and capacitive properties. I am sure that this study will help in understanding and developing new composites for energy storage in the future.

Thank you very much!

The manuscript can be considered for publications in Membranes after tackling out these two comments:

1. The authors should convey more recent literatures on Iron/Graphene and/or carbon materials in the introduction for superconductors and lithium-ion batteries. For example: https://www.mdpi.com/2079-4991/9/5/776

Please, find recent literature 1-4, 12-14, 16, 17 in References. The paper suggested by Referee was also added.

2. I wonder if the authors can study more samples with different Fe3O4/G mass ratio, e.g. 3:19 and 4:1.

Yes, of course! For this goal we should change the dimensions of graphene or number of magnetite unit cells.

Round 2

Reviewer 2 Report

The revised version of the work entitled Graphene/Fe3O4 nanocomposite as a promising material for chemical current sources: a theoretical study by Shunaev and Glukhova have performed cosmetic changes after this first revision.

 The manuscript in the present form still suffers from the required novelty and potential impact to be published. Consequently, I don´t recommend to publish this work.

  1. The main concern of this work is the unrealistic system under study. The main challenge to use graphene in supercapacitors or batteries is to functionalize it preserving its electronic properties. Functionalization is usually needed in order to prevent the agglomeration of the active parts, such as nanoparticles, with the subsequent decrease of performance due to the decrease of surface-to-volume ratio. The present work is unrealistic since the authors study non oxidized graphene with clusters of around 1 nm size. Strikingly, they compare with the literature where the crystal size of the Fe3O4 studied cover the range of several tens of nanometers, i.e. between one and two orders of magnitude higher than the present work. See for instance references 16 and 17. Furthermore, the reference 16 presents graphene sheets containing oxidized species as derived from the XPS results presented.
  2. The authors perform calculations in free standing. How realistic are sheets of graphene of only 2 to 6 nm free standing in supercaps and batteries?

Other comments:

  1. From figure 1, the cluster of Fe3O4 considered looks the very same in all the cases under study. Are the clusters relaxed or considered unaltered in the simulations against G-Fe3O4 distances? How realistic is to consider a particular registry between G and Fe3O4 particles?
  2. The electrochemical activity of particles depends not only in their size but also their crystallographic plane orientations. See for instance experimental work of reference 17 which determined a cubic phase. What’s the structure of the particle considered in this work?
  3. The authors should clarify if the Fermi level shift is due to the presence of the Fe3O4 particle or just due to the bending of graphene.

In conclusion, considering all the comments above, I am inclined to reject the present work for publication.

Author Response

  1. The main concern of this work is the unrealistic system under study. The main challenge to use graphene in supercapacitors or batteries is to functionalize it preserving its electronic properties. Functionalization is usually needed in order to prevent the agglomeration of the active parts, such as nanoparticles, with the subsequent decrease of performance due to the decrease of surface-to-volume ratio. The present work is unrealistic since the authors study non oxidized graphene with clusters of around 1 nm size. Strikingly, they compare with the literature where the crystal size of the Fe3O4 studied cover the range of several tens of nanometers, i.e. between one and two orders of magnitude higher than the present work. See for instance references 16 and 17. Furthermore, the reference 16 presents graphene sheets containing oxidized species as derived from the XPS results presented.

Why should we consider non-oxidized graphene? We gave several Refs where Fe3O4 nanoparticles are attached to pure graphene (even non rGO).

We agree that size of iron nanoparticle in real experiment is much higher than in our study. But every modeling suggests simplifications since computer resources are limited by amount of considered atoms. Note that we applied quantum chemical SCC DFTB method that requires many resources since it takes into account electronic interaction and charge transfer. So the main idea of the paper was to consider nanocomposite with different mass ratios as in real experiment.

  1. The authors perform calculations in free standing. How realistic are sheets of graphene of only 2 to 6 nm free standing in supercaps and batteries?

There are many papers where free standings graphene films are used as materials for batteries. For example, https://doi.org/10.1016/j.elecom.2010.08.008 («Flexible free-standing graphene-silicon composite film for lithium-ion batteries»), https://doi.org/10.1021/acsami.8b10066 (Flexible free-standing graphene-silicon composite film for lithium-ion batteries), https://doi.org/10.1021/am500996c (Highly Conductive Freestanding Graphene Films as Anode Current Collectors for Flexible Lithium-Ion Batteries), etc. Note that we considered not finite graphene sheets. 2 to 6 nm is the dimensions of supercell that is located in periodic box.

  1. From figure 1, the cluster of Fe3O4 considered looks the very same in all the cases under study. Are the clusters relaxed or considered unaltered in the simulations against G-Fe3O4 distances? How realistic is to consider a particular registry between G and Fe3O4 particles?

Of course, all clusters were relaxed in periodic box as it was written in manuscript. Since iron nanoparticles interacted with only graphene skeleton they had similar form after simulation.

When the distance between graphene and iron nanoparticle was changed only graphene atoms were optimized. When iron nanoparticles were too far from graphene sheet it was flat. When iron nanoparticles reached graphene it became curve.

  1. The electrochemical activity of particles depends not only in their size but also their crystallographic plane orientations. See for instance experimental work of reference 17 which determined a cubic phase. What’s the structure of the particle considered in this work?

Yes, it’s right. We used to the cubic space group Fd3m as it was in Ref. 17. The structure was taken from https://www.materialsproject.org/materials/mp-19306/. We add this data to the paper

  1. The authors should clarify if the Fermi level shift is due to the presence of the Fe3O4 particle or just due to the bending of graphene.

The phrase was added.

«It is noticeable that the doping of graphene with magnetite particles shifts the Fermi level to zero due to the presence of graphene oxide, as in the case of CNT doping with maghemite particles [21].»

Reviewer 3 Report

It would be more perfect if there is a small amount of experimental data support

Author Response

Our study is theoretical but it is based on experiment.

  • The iron nanoparticle atomic structure had the cubic space group Fd3m as in [17].
  • Objects of the study were the 2D-composites with mass ratios m(Fe3O4):m(G) = 1:9, 1:4, 3:7 и 1:1, since these concentrations can clearly demonstrate gradual change of graphene capacitive properties with addition of magnetite nanocomposites and such composites can be synthesized in experiments [16, 17, 22].
  • These observations may be useful during design the electrodes for flexible storage devices on the base of G/Fe3O4 composites [7-11, 16, 17].

We note that these results were obtained in the frame of grant that is held with experimenters. So we are looking forward to support of data by experiment in future.